# Sorry Parents, Children Consume High Amounts of Candy before and after a Meal: Within-Person Comparisons of Children’s Candy Intake and Associations with Temperament and Appetite

**DOI:** 10.3390/children10010052

**Published:** 2022-12-26

**Authors:** Erika Hernandez, Amy M. Moore, Brandi Y. Rollins, Alison Tovar, Jennifer S. Savage

**Affiliations:** 1Center for Childhood Obesity Research, 129 Noll Laboratory, The Pennsylvania State University, University Park, PA 16802, USA; 2Biobehavioral Health, The Pennsylvania State University, 118 Henderson Building, University Park, PA 16802, USA; 3Department of Behavioral and Social Sciences, Brown University School of Public Health, Box G-S121-4, Providence, RI 02912, USA; 4Department of Nutritional Sciences, 110 Chandlee Laboratory, The Pennsylvania State University, University Park, PA 16802, USA

**Keywords:** eating in the absence of hunger, temperament, appetite, candy

## Abstract

Candy provides little nutritional value and contributes to children’s energy intake from added sugars. Factors influencing children’s candy intake remain largely unknown. This study describes children’s total candy intake (kcal) before and after a meal and examines associations of candy intake in both conditions with children’s temperament and appetite among a predominantly White, highly educated sample. Children (*n* = 38, age 5–8 years) were given free access to 11 candies (5 chocolate, 6 non-chocolate) and non-food alternatives during a pre-meal and a post-meal condition. Parents completed the Child Behavior Questionnaire and the Child Eating Behavior Questionnaire. Total candy intake was less when offered after a meal (209.3 kcal; SD = 111.25) than before a meal when still hungry (283.6 kcal; SD = 167.3), but not statistically different. Individual differences in candy intake between conditions was calculated to categorize children into three groups: “Better Regulators” consumed more candy before a meal (39%), “Consistent/Poorer Regulators” consumed similar amounts before and after a meal regardless of hunger (32%), and “Most Disinhibited” children consumed more candy after a meal when not hungry (29%). The “Better Regulators” group was lowest in negative affect and the “Consistent/Poorer Regulators” group was highest in food responsiveness. Children’s candy intake was high relative to daily energy needs both before and after a meal. Child negative affect and food responsiveness appear to be child characteristics that predispose children to poor self-regulation of candy intake before and after a meal.

## 1. Introduction

Excess intake of sweets (e.g., desserts, candy, and sweetened beverages) may lead to negative physical health outcomes [1,2] including overweight [3] and dental caries [4]. Candy in particular provides little nutritional value and is a contributor to children’s energy intake from added sugars, contributing approximately 5–9% of children’s daily energy intake from added sugars [5], which exceeds the Dietary Guidelines [6]. Despite the lack of nutritional value, one-third of US children consume candy each day [7], highlighting the need for research identifying factors that influence children’s candy intake. While current efforts to modify the food environment and make candy less accessible and less marketed are critical and underway, it is also important to understand factors that influence candy intake within the home environment. One such factor is the timing of candy intake around a meal in the home; qualitative evidence suggests that parents believe that offering children candy before a meal may displace intake of “real foods” [8], yet little is known about children’s candy intake before and after a meal. There is currently no published guidance for parents regarding timing of serving sweets to children. Further, children’s individual differences such as temperament and appetite are associated with intake of highly palatable foods [9,10], but little is known about how these individual differences are associated with candy intake before and after a meal. In the current study, we used a within-person design to examine children’s candy intake in both a pre-meal and post-meal condition. Understanding the influence of serving timing around a meal as well as individual differences on children’s candy intake can help tailor future interventions to reduce the intake of highly palatable foods in the home environment, such as candy.

Children’s food preferences and eating behaviors are shaped by biological and environmental factors [11,12] and may also influence candy intake. Eating behaviors, such as higher food responsiveness and lower satiety responsiveness, are associated with child overweight [9,13]. Eating in the absence of hunger (EAH), an indicator of food responsiveness, is a component of food intake self-regulation that refers to children’s tendency to consume highly palatable foods despite being satiated [14]. Classically, EAH is a laboratory-based protocol where children are provided an ad libitum standardized meal, then a hunger assessment is completed, followed by free access to a variety of palatable snack foods and non-food alternatives such as toys [15]. Although there are large interindividual differences in how many calories children consume during the EAH protocol, greater energy intake from palatable foods is consistently associated with increased adiposity across childhood [16]. Pertinent for the current study, there may be individual differences in how children respond to food cues when candy is available before and after a meal, such as with child temperament.

Child temperament refers to individual differences in reactivity and self-regulation [17]. Temperament has emerged as a trait that influences children’s dietary behaviors and weight status [18] and thus may influence candy intake. The three broad dimensions of temperament include negative affect (characterized by mood instability and dysregulated negative emotions), effortful control (characterized by refraining from a desired behavior while staying on task), and surgency (characterized by high impulsivity and pleasure-seeking). Temperament is broadly reflective of how children approach their environments, and as such, this may carry over into other contexts including self-regulation of eating and food choices [19]. One study conducted with preschool-aged children from low-income households found that children higher in surgency or lower in negative affectivity consume more energy during the EAH task [10]. Understanding the impact of child temperament on food intake, specifically palatable foods that are high in added sugars such as candy, is important to help create targeted interventions by identifying children who may be most susceptible to high intake of high energy-dense foods.

The overall aim of this within-person study was to describe children’s candy intake when served before versus after a meal, and given previous literature on children’s liking and craving of chocolate [20,21,22], to further characterize intake by candy type (chocolate and non-chocolate candy). We hypothesized that children would consume more candy before a meal when still hungry. The secondary aim was to understand how individual characteristics such as child temperament and appetitive traits were associated with candy intake (kcal) in both conditions. Based on previous findings from the EAH literature, we hypothesized that children higher in surgency and satiety responsiveness, but lower in negative affect and food responsiveness would show poorer self-regulation of candy intake.

## 2. Materials and Methods

### 2.1. Study Design & Participants

This study was a secondary analysis of data from an observational study, designed to explore food parenting and factors that influence child candy intake among parents with children from 5 to 8 years of age (*n* = 43). Recruitment occurred in central Pennsylvania through flyers posted at local schools and website advertisements. Parents were eligible if their child was between 5 to 8 years of age and consumed candy at least once per month. Parents were excluded if their child had a medical condition that impacted their food intake or if the child had known food allergies. Eligible parents were enrolled and asked to provide written informed consent and parental permission for their children. The Pennsylvania State University Institutional Review Board approved all study procedures (protocol #PRAMS00042379).

### 2.2. Procedure

Parents and their children visited the laboratory for two separate 2-h visits. Session 1 assessed children’s candy intake after a meal. Parents were instructed to provide breakfast at home and a standardized lunch was provided. Children also completed a hunger and food preference assessment with research staff and participated in the standard EAH protocol. While children participated in the EAH protocol, parents completed paper surveys. Survey responses were double entered by research assistants and any discrepancies were reviewed and resolved. Session 2 assessed children’s candy intake before a meal. In this condition, children ate a standardized breakfast followed by a 2-h break. Next, children completed a hunger and food preference assessment with research staff and then participated in the EAH protocol. This session occurred 2–3 months after session 1. See Figure 1.

### 2.3. Protocols

#### 2.3.1. Post-Meal Candy Intake

In the post-meal condition, children were served a standard ad libitum lunch to minimize the influence of hunger on children’s intake of candy. Each child self-selected either a turkey (128 g), ham (128 g), or cheese (82 g) sandwich, and were provided with cheese slices (11 g), mayonnaise (13 g), mustard (5 g), apple slices (68 g), and low-fat milk (12 oz). Lunch was approximately 545–595 kcal, depending on sandwich selection. Children were given 20 min to eat. Next, a hunger assessment was administered to measure children’s hunger and fullness using figures that depicted an empty stomach (“hungry”), a half-full stomach (“half-full”), and a full stomach (“full”). After the hunger assessment, a food preference assessment was conducted to measure children’s preference for the 11 candies included in the EAH protocol. Next, children were shown various toys (e.g., blocks, colorings books) that were available for them to play with during the protocol. Generous portions of the 11 candies from the preference assessment were provided: SweeTarts (50 g), Haribo Gold-Bears Gummi Candy (100 g), Skittles (75 g), Swedish Fish (75 g), M&Ms (75 g), Hershey’s Nuggets (67 g), Snickers Miniatures candy bars (61 g), Kit Kats Miniatures chocolate bars (61 g), Reese’s Peanut Butter Cups miniatures (61 g), Goetze’s Caramel Creams (71 g), and Dum Dum Lollipops (28 g). Children were instructed that they could eat any of the foods or play with the toys while the researcher worked in the adjacent room. The researcher then left the room for 7 min. Total energy (kcal) from candy was calculated by pre- and post-weighing all foods and manufacturers’ information was used to convert gram weight consumed into total energy consumed (kcal) across all 11 candies.

#### 2.3.2. Pre-Meal Candy Intake

In the pre-meal condition, children were served a standard ad libitum breakfast and had a 2-h break before completing the EAH protocol before lunch. Each child self-selected from blueberry or snickerdoodle mini muffins (24 g), peanut butter or cinnamon raisin granola bars (24 g), five breakfast cereals (18–35 g), a fruit cup (113 g), and skim milk (8 oz). Breakfast was approximately 400–440 kcal, depending on food selection. Children were given 20 min to eat. The same hunger and food preference assessments were administered and children were given access to the same 11 candies and non-food alternatives for 7 min. Total energy (kcal) from candy consumed was calculated in the same way as the post-meal condition.

### 2.4. Measures

#### 2.4.1. Temperament

Child temperament was measured using the 36-item Children’s Behavior Questionnaire (CBQ-VSF) [23]. The current study examines three broad dimensions (i.e., superfactors): negative affect (12 items; α = 0.61, e.g., “Is very difficult to soothe when s/he has become upset”), effortful control (10 items; α = 0.51, e.g., “Is good at following instructions”), and surgency (12 items; α = 0.73, e.g., “Often rushes into new situations”). Two items were removed from effortful control because of negative correlations with the rest of the scale [24]. Parents reported their child’s reaction to certain situations within the past six months. Items were scored on a scale from 1 (extremely untrue of your child) to 7 (extremely true of your child), with higher mean scores indicating greater levels of that temperament dimension.

#### 2.4.2. Appetitive Traits

Children’s appetitive traits were measured using the 35-item Child Eating Behavior Questionnaire (CEBQ) [25]. Specifically, food responsiveness (5 items; α = 0.73, e.g., “Even if my child is full up s/he finds room to eat his/her favorite food”) and satiety responsiveness (5 items; α = 0.65, e.g., “My child gets full easily”) were included in the current study. Items were scored on a scale from 1 (never) to 5 (always), with higher mean scores indicating greater levels of that eating behavior.

### 2.5. Statistical Analyses

All analyses were conducted in SAS 9.4 (SAS Institute, Cary, NC, USA). Descriptive statistics were computed and normality was assessed for study variables. Following standard procedures, children who responded that they were “hungry” during the post-meal condition were removed from the current analyses (*n* = 5). Thus, the analytic sample for the current paper is *n* = 38.

General linear models were conducted to examine children’s energy intake from candy in both the pre-meal and post-meal conditions. Additional models for chocolate and non-chocolate candy intake were conducted separately. To test associations between children’s candy intake with temperament and appetite, a difference score was calculated by subtracting the amount of kcal children consumed during the pre-meal condition from the amount of kcal children consumed during the post-meal condition. Difference scores for children’s candy kcal intake in both the pre-meal and post-meal conditions were then categorized into three groups for ease of interpretability of these difference scores. ANOVAs examined associations between candy intake group membership and child temperament dimensions (negative affect, effortful control, and surgency) as well as appetitive traits (food responsiveness, satiety responsiveness). Models were conducted separately for each temperament and appetitive trait outcome. Given the relatively small sample size, multiple comparisons were examined for models *p* < 0.10. For all analyses, child sex, child age, and annual household income were tested as covariates but were not retained because none reached statistical significance at *p* < 0.05.

## 3. Results

### 3.1. Participant Characteristics

Parents self-identified as mothers (97%), White (95%), non-Hispanic (97%), married (87%), and well-educated (40% college graduates, 48% graduate degree). Children were on average, 7.2 years old, male (50%), White (92%), and non-Hispanic (96%). During the post-meal condition, 71.1% of children were “half full” and 28.9% were “full”. During the pre-meal condition, 30.2% of children were “half full” and 69.8% were “hungry”.

### 3.2. Candy Intake before and after a Meal

There was no significant difference in the amount of total candy children consumed between conditions: post-meal (mean = 209.3 kcal; SD = 111.25; range = 48.3–463.7 kcal) and pre-meal (mean = 283.6 kcal; SD = 167.3; range = 2.3–786.5 kcal) (*p* = 0.15). There was also no significant difference in the amount of chocolate candy children consumed between conditions: post-meal (mean = 109.7 kcal; SD = 100.25; range = 0.0–386.0 kcal) and pre-meal (mean = 156.6 kcal; SD = 135.1; range = 0.0–653.1 kcal) (*p* = 0.78). However, children consumed significantly less non-chocolate candy during the post-meal condition (mean = 99.5 kcal; SD = 72.4; range = 0.0–331.3 kcal) than the pre-meal condition (mean = 127.1 kcal; SD = 120.0; range = 0.7–413.9 kcal) (*p* = 0.05). Figure 2 shows candy intake by condition.

### 3.3. Differences in Candy Intake between Conditions and Associations with Temperament and Appetite

On average, children consumed a difference of 74.3 kcal (SD = 135.1) between the pre- and post-meal conditions; consuming more before a meal. This difference score was then used to categorize children’s candy intake between conditions for interpretability. The first group, which we term the “Better Regulators”, included children who consumed more before a meal when hungry than after a meal when no longer hungry (*n* = 15; 39%; range of difference scores pre/post meal = 102.8 kcal–475.1 kcal). The second group, which we term “Consistent/Poorer Regulators”, included children who consumed similar amounts of candy before and after a meal (*n* = 12; 32%; range of difference scores pre/post meal = 4.7 kcal–91.8 kcal). The third group, which we term “Most Disinhibited”, included children who consumed more after a meal, despite not being hungry (*n* = 11; 29%; range of difference scores pre/post meal = −10.5 kcal–−241.9 kcal). Candy intake (kcal) cutoffs for grouping was a data-driven decision that sought to create balanced group sizes. Children’s candy intake group was marginally associated with their negative affect (*p* = 0.07). Multiple comparisons indicated that the “Better Regulators” were significantly lower on negative affect than the “Consistent/Poorer Regulators” (*p* = 0.03) and “Most Disinhibited” (*p* = 0.03) children. See Figure 3. Children’s candy intake group membership was also related to food responsiveness (*p* = 0.03). Children in the “Consistent/Poorer Regulators” group were marginally higher on food responsiveness than the “Better Regulators” (*p* = 0.06) and higher than the “Most Disinhibited” children (*p* = 0.02). See Figure 4. There was no significant association between candy intake group and surgency, effortful control, or satiety responsiveness. See Table 1.

## 4. Discussion

This analysis described children’s candy intake during pre- and post-meal conditions and examined associations with child temperament (individual differences in reactivity and self-regulation) and appetite (responsiveness to external food cues and internal hunger cues) among predominantly White children 5 to 8 years of age from highly educated families. Candy has little nutritional value, yet children’s candy intake was high relative to daily energy needs [6], regardless of timing before or after a meal. On average, children in our sample only ate about 75 fewer kcal when candy was offered post-meal, with about 40% of children eating more before a meal. Our data also show that high child negative affect and food responsiveness may increase child susceptibility to poor self-regulation of candy intake before a meal, when hungry, and after a meal, when no longer hungry.

Childhood obesity is the result of a daily energy imbalance, with the energy gap estimated somewhere between 30–165 kcal per day [14,26]. Given the high energy density of candy and the little nutritional value it offers, candy may be an important target for reducing children’s energy gap in the home environment that contributes to the high rates of childhood obesity in the US [27]. Our data show that children did not eat statistically less candy in the post-meal (209 kcal) when not hungry than the pre-meal (284 kcal). However, data reveal that children ate significantly less non-chocolate candy after a meal than before when hungry, but similar amounts of chocolate candy by condition. One possible explanation for this finding is that chocolate is often regarded as one of the most palatable, liked [20], and craved foods in Western societies [21], and as one of children’s most liked foods [22]. Thus, children may be more likely to override their internal hunger/fullness cues to consume chocolate candy in particular, both before and after a meal.

Using a within-person design, we also tested individual differences in candy intake by creating a difference score in candy intake by condition to understand if children ate similar amounts regardless of the timing of a meal or not. We anticipated that children would eat more candy when still hungry and offered before a meal than after a meal when not hungry. However, this only occurred for ~40% of children; one group of children illustrated the best self-regulation of candy intake by compensating, or eating less candy in the post-meal than pre-meal condition [28,29]. The other two groups, which were the majority of children (60%), ate a similar amount of candy regardless of condition and/or more candy after a meal when not hungry than before a meal. As such, our data indicate that parents should not expect all children to be able to rely on their hunger/fullness cues to regulate their own intake of candy. Some children likely need more monitoring than others to regulate intake of this highly palatable food. However, given that parents’ restriction of highly palatable foods often has the unintended effect of increased liking of the restricted food [30], parents should be encouraged to establish clear and consistent rules and routines around food that children can anticipate (i.e., structure-based feeding practices), and to limit access to more energy dense foods by not bringing these foods into the home [31].

Previous research suggests that children’s external eating [32], surgency, and negative affect [10], are associated with energy intake during EAH. Building on this literature, we explored how child characteristics (e.g., temperament and appetite) increase child susceptibility to poor self-regulation of candy intake before and after a meal. Recent literature has begun to recognize the interplay between children’s developing general self-regulation and self-regulation of food intake [19]. Our data indicate that negative affect was associated with children’s candy intake, such that children in the “Better Regulators” group were the lowest on negative affect. Negative affect, which is characterized by mood instability and dysregulated negative emotions [17], is considered an aspect of children’s emotion regulation. In contrast to our findings, previous research has found that children low in negative affect tend to consume higher total energy during EAH [10]. Differences in protocols between these studies may help explain our divergent findings; Leung and colleague’s paper utilized a standard EAH protocol after a meal with both sweet and salty food options, whereas the current study examined differences in intake before and after a meal of candy only. Further, negative affect has also been linked to disordered eating in adults [33] and higher obesity risk for children [18], suggesting some overlap between these domains of self-regulation and thus implications for improving children’s ability to self-regulate intake.

Lastly, in the current study, food responsiveness was also associated with children’s candy intake, such that children who had similar candy intake by condition, regardless of hunger, were the highest on food responsiveness. Food responsiveness refers to children’s tendency to eat in response to external food cues, such as the sight of food [14]. Thus, it is logical that the “Consistent/Poorer Regulators” group ate similar amounts of candy before and after a meal, because this group of children tends to respond to external food cues rather than internal hunger and fullness cues. Sensory specific satiety may also help explain why children ate similar amounts of candy before and after a meal [34,35]. For some children, candy offered after a meal is even more desirable, leading to overriding hunger/fullness cues and consumption of a similar amount of candy after a meal than before a meal. However, it would have been expected that children in the “Most Disinhibited” group were highest on food responsiveness, as children in this group ate more candy after a meal than before. Our findings suggest that promoting children’s ability to tune into their own hunger and fullness cues, and thus away from external cues, may be an effective strategy to promote children’s ability to regulate their intake of candy, a contributor of energy intake from added sugars in children’s diets.

### Limitations and Strengths

The primary limitation of the current study is the small sample size, which limited the analyses we were able to conduct in the current paper. Further, these data were collected from a predominantly White, highly educated sample, thus limiting the generalizability of findings. As such, future research should seek to expand these findings to families underrepresented in the literature, such as those from racially and ethnically diverse communities as well as low-income backgrounds. A strength of the current study was the within-subjects design in examining total energy (kcal) from candy intake in both a pre-meal and post-meal condition, as well as a focus on candy exclusively in these paradigms. Both are novel contributions to the field. However, the pre- and post-meal conditions were not counterbalanced (though there was a 2–3-month wash-out period between conditions) and breakfast was served during one condition only, which are additional limitations. Future research in this area should be conducted on a larger, more diverse sample, with a counterbalanced design to identify robust associations between child temperament, appetite, and energy intake from candy before and after a meal, as well as explore patterns by candy type and child sex. Longitudinal studies will also be critical in the next steps of this research, as the age range studied here spanned multiple years of childhood, limiting our understanding of developmental process.

## 5. Conclusions

We assessed energy intake of candy in both pre- and post-meal conditions and identified that children’s candy intake was high relative to daily energy needs regardless of timing around a meal. Children consumed high amounts of chocolate candy in particular. We classified children’s candy intake based on the difference in kcal intake between conditions and found that the majority of children were poorly regulated, eating a similar amount of candy before and after a meal or more after a meal. Further, child characteristics including negative affect and food responsiveness may predispose children to poor self-regulation of candy intake. Highly palatable foods such as candy likely override children’s hunger and fullness cues, suggesting the need for parents to provide additional support to help children’s developing self-regulation of intake in the home environment. Parents of children high in negative affect and food responsiveness will likely need additional messaging to promote children’s healthy self-regulation of candy intake.

## Figures and Tables

**Figure 1 children-10-00052-f001:**
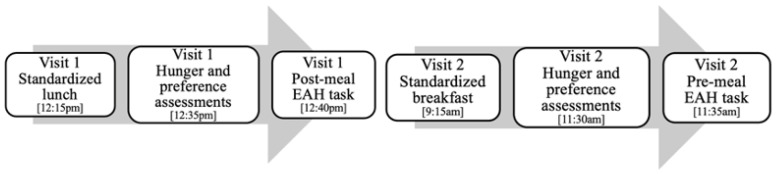
Visual depiction of study protocol.

**Figure 2 children-10-00052-f002:**
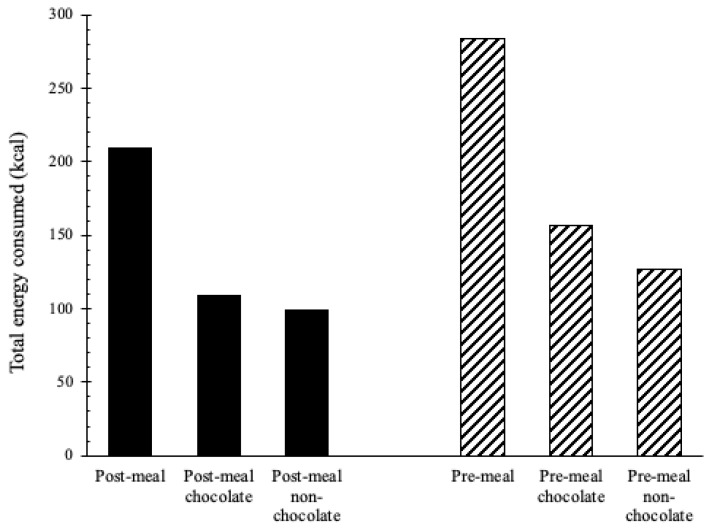
Children’s total energy (kcal) from candy intake during the pre- and post-meal conditions by child sex and candy type (*n* = 38). Chocolate candies = M&Ms, Hershey’s Nuggets, Snickers Miniatures candy bars, Kit Kats Miniatures chocolate bars, Reese’s Peanut Butter Cups miniatures. Non-chocolate candies = SweeTarts, Haribo Gold-Bears Gummi Candy, Skittles, Swedish Fish, Goetze’s Caramel Creams, and Dum Dum Lollipops.

**Figure 3 children-10-00052-f003:**
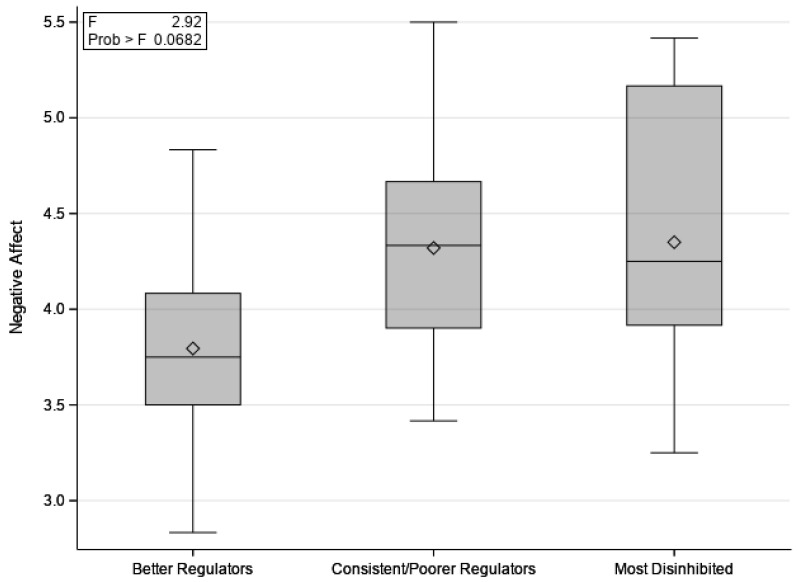
Means of children’s negative affect for “Better Regulators” (*n* = 15), “Consistent/Poorer Regulators” (*n* = 12), and “Most Disinhibited” (*n* = 11) groups.

**Figure 4 children-10-00052-f004:**
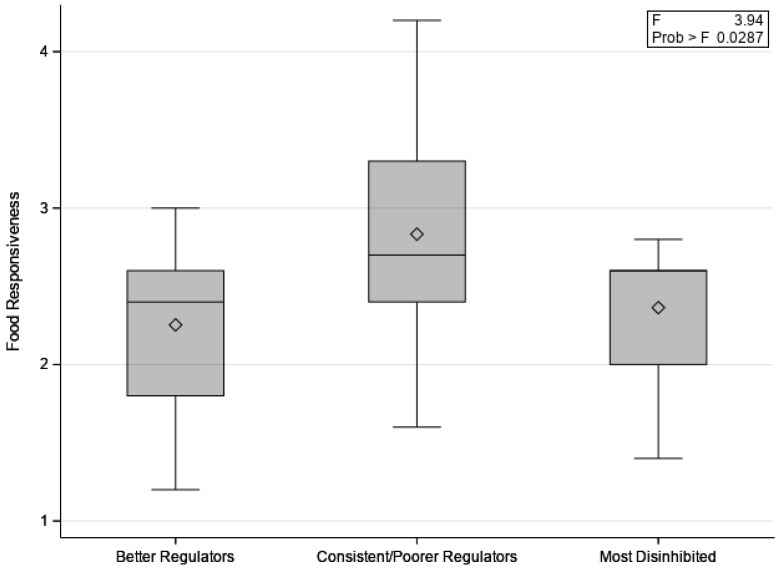
Means of children’s food responsiveness for “Better Regulators” (*n* = 15), “Consistent/Poorer Regulators” (*n* = 12), and “Most Disinhibited” (*n* = 11) groups.

**Table 1 children-10-00052-t001:** Results examining associations between children’s candy intake group with temperament and appetite.

	*F* Value	*p* Value	R^2^
Negative Affect	2.92	0.07	0.15
Effortful Control	1.84	0.18	0.10
Surgency	1.77	0.19	0.10
Food Responsiveness	3.94	0.03	0.18
Satiety Responsiveness	2.08	0.14	0.11

## Data Availability

The data presented in this study are available on request from the corresponding author. The data are not publicly available due to privacy or ethical restrictions.

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
