# Peer review of "Sorry Parents, Children Consume High Amounts of Candy before and after a Meal: Within-Person Comparisons of Children’s Candy Intake and Associations with Temperament and Appetite"

_children, 2022, doi:10.3390/children10010052_

Round 1

Reviewer 1 Report

Overall the study methodology is largely appropriate for examining the effects of meals on candy intake in children.  However the complete lack of diversity of the population (lines 190ff " Parents self-identified as mothers (97%), White (95%), non-Hispanic (97%), married 190 (87%), and well-educated (40% college graduates, 48% graduate degree) limits the applicability of the study.  This should be mentioned in either the title or the abstract – Sorry White Highly Educated Parents…. or at the least qualify it in the abstract.

 The authors open the abstract with a somewhat misleading statement on line 15-16 that candy is a "leading" source of added sugars in children.  In fact candy only contributes between 5-10% of added sugars to children's diet with sugar sweetened beverages, sweet deserts and sweetened yogurt contributing a far greater percentage of added sugars (see Herrick et al. and Dietary Guidelines for Americans1,2).  This does not invalidate the concern over candy consumption but should be corrected to reflect actual intake levels.

This error is repeated on lines 36-37 where the authors misquote Bailey et al. saying that candy contributes 14% of added sugars when in fact that article states that 14% of calories in children comes from all sources of added sugars while candy consumption is far less and only about 5% of this 14% thus being less than 1% of total intake.    Candy certainly is a contributor to added sugar intake and it is reasonable to highlight and study but mischaracterizing the intake to make it seem a greater problem than it actually is needs correcting.

Line 17-18:  Clarity is needed "This study describes children’s total candy intake (kcal) before and after a meal, and tests individual 17 differences in children’s candy intake with child temperament and appetite."  Do the authors mean …. when correlated with child temperament? … in temperament and appetite with respect to candy intake?

The introduction should be expanded.  It currently focuses on temperament and does not discuss liking for sweet taste, dietary restraint, other personality characteristics beyond temperament.  While it is fine to focus on this aspect other possible influences should be included in introducing the topic and why temperament was chosen as the aspect to focus on should be justified. 

The methodology is well described.

The discussion again suffers from overstating the importance of candy as a contributor to caloric intake and there needs to be more qualification of the applicability of the results to other than highly educated, white, high SES families.

1.         US Department of Agriculture, Services UDoHaH. Dietary Guidelines for Americans, 2020-2025 2020.

2.         Herrick KA, Fryar CD, Hamner HC, Park S, Ogden CL. Added Sugars Intake among US Infants and Toddlers. Journal of the Academy of Nutrition and Dietetics. 2020;120(1):23-32.

Author Response

Reviewer comment

Response

Location information

Overall the study methodology is largely appropriate for examining the effects of meals on candy intake in children. However the complete lack of diversity of the population ("Parents self-identified as mothers (97%), White (95%), non-Hispanic (97%), married 190 (87%), and well-educated (40% college graduates, 48% graduate degree) limits the applicability of the study.  This should be mentioned in either the title or the abstract – Sorry White Highly Educated Parents…. or at the least qualify it in the abstract.

We thank the reviewer for their comment on our methodology. We agree that the lack of diversity limits the applicability of the results. We have added this contextual information when discussing our sample into the abstract and discussion.

“This study describes children’s total candy intake (kcal) before and after a meal and examines associations of candy intake in both conditions with children’s temperament and appetite among a predominantly White, highly educated sample.”

“Further, these data were collected from a predominantly White, highly educated sample, thus limiting the generalizability of findings. As such, future research should seek to expand these findings to families underrepresented in the literature, such as those from racially and ethnically diverse communities as well as low-income backgrounds.”

Abstract, lines 18-19

Discussion, lines 348-349 and 423-426

The authors open the abstract with a somewhat misleading statement on line 15-16 that candy is a "leading" source of added sugars in children.  In fact candy only contributes between 5-10% of added sugars to children's diet with sugar sweetened beverages, sweet deserts and sweetened yogurt contributing a far greater percentage of added sugars (see Herrick et al. and Dietary Guidelines for Americans1,2).  This does not invalidate the concern over candy consumption but should be corrected to reflect actual intake levels.

This sentence in the abstract has been amended to reflect the reviewer’s comment.

“Candy provides little nutritional value and contributes to children’s energy intake from added sugars.”

Abstract, line 15

This error is repeated on lines 36-37 where the authors misquote Bailey et al. saying that candy contributes 14% of added sugars when in fact that article states that 14% of calories in children comes from all sources of added sugars while candy consumption is far less and only about 5% of this 14% thus being less than 1% of total intake.    Candy certainly is a contributor to added sugar intake and it is reasonable to highlight and study but mischaracterizing the intake to make it seem a greater problem than it actually is needs correcting.

These sentences in the introduction have also been amended to reflect the reviewer’s comment.

“Candy in particular provides little nutritional value and is a contributor to children’s energy intake from added sugars, contributing approximately 5-9% of children’s daily energy intake from added sugars [5], which exceeds the Dietary Guidelines for added sugar [6].”

Introduction, lines 37-39

Line 17-18:  Clarity is needed "This study describes children’s total candy intake (kcal) before and after a meal, and tests individual 17 differences in children’s candy intake with child temperament and appetite."  Do the authors mean …. when correlated with child temperament? … in temperament and appetite with respect to candy intake?

We have clarified this sentence.

“This study describes children’s total candy intake (kcal) before and after a meal and examines associations of candy intake in both conditions with children’s temperament and appetite among a predominantly White, highly educated sample.”

Introduction, lines 17-19

The introduction should be expanded.  It currently focuses on temperament and does not discuss liking for sweet taste, dietary restraint, other personality characteristics beyond temperament.  While it is fine to focus on this aspect other possible influences should be included in introducing the topic and why temperament was chosen as the aspect to focus on should be justified. 

We thank the reviewer for their comment. We chose to examine child temperament because of the research indicating associations between children’s appetite and temperament with intake of highly palatable foods, suggesting the importance of studying these individual differences when examining candy intake. We have edited the first paragraph of the introduction to help foreshadow the temperament paragraph.

“Further, children’s individual differences such as temperament and appetite are associated with intake of highly palatable foods [9,10], but little is known about how these individual differences are associated with candy intake before and after a meal. In the current study, we used a within-person design to examine children’s candy intake in both a pre-meal and post-meal condition. Understanding the influence of serving timing around a meal as well as individual differences on children’s candy intake can help tailor future interventions to reduce the intake of highly palatable foods in the home environment, such as candy.”

Introduction, lines 56-61

The methodology is well described.

We once again thank the reviewer for the positive comments on our methodology.

N/A

The discussion again suffers from overstating the importance of candy as a contributor to caloric intake and there needs to be more qualification of the applicability of the results to other than highly educated, white, high SES families.

We have edited the discussion to reflect this comment.

“Our findings suggest that promoting children’s ability to tune into their own hunger and fullness cues, and thus away from external cues, may be an effective strategy to promote children’s ability to regulate their intake of candy, a contributor of energy intake from added sugar in children’s diets.”

“Further, these data were collected from a predominantly White, highly educated sample, thus limiting the generalizability of findings. As such, future research should seek to expand these findings to families underrepresented in the literature, such as those from racially and ethnically diverse communities as well as low-income backgrounds.”

Discussion, lines 417-420, 348-349, and 423-426

Reviewer 2 Report

This is a survey study including 38 participants. Various topics of chocolate were recorded and compared before and after meal. They found that Children’s candy intake was high relative to daily energy needs both before and after a meal.

This is an interesting study with some new findings in this area of research. The sample size of subjects is samll for analysis. However, I nevertheless have the following comments that required to be addressed.

1.     The statistical methods used and described should be made specific to the research question. Professional editing for improvement of “Statistical analysis” is suggested.

2.     For tables, I suggested to add a table about results of ANOVA.

3.     Please add a flow chart to increase readability.

4.     Please brief the conclusions to add readability.

5.     Any study involved health theory for this issue?

Author Response

Reviewer comment

Response

Location information

The statistical methods used and described should be made specific to the research question. Professional editing for improvement of “Statistical analysis” is suggested.

We have edited the Statistical Analysis paragraph for readability.

Method, lines 196-218

For tables, I suggested to add a table about results of ANOVA.

Table 1 has been added to the results to reflect the results showing the association between children’s candy intake group with temperament and appetite.

Results, line 342

Please add a flow chart to increase readability.

As suggested, a visual depiction of the study protocol has been added to the method section to increase readability.

Method, line 126

Please brief the conclusions to add readability.

We have shortened the Conclusions paragraph.

Discussion, lines 440-452

Any study involved health theory for this issue?

We thank the reviewer for this comment. We did not have a specific theoretical framework for the current study, though our research questions were guided by the emerging conceptualization of food and non-food related self-regulation, which is included in the discussion.

Discussion, lines 391-405

Round 2

Reviewer 2 Report

No further comments.